# Mechanosensitive Aspects of Cell Biology in Manual Scar Therapy for Deep Dermal Defects

**DOI:** 10.3390/ijms21062055

**Published:** 2020-03-17

**Authors:** Thomas Koller

**Affiliations:** Musculoskeletal Physiotherapy, Orthopaedic and Hand Surgery Rehabilitation, Bellikon Rehabilitation Clinic, Mutschellenstrasse 2, 5454 Bellikon, Switzerland; thomas.koller@rehabellikon.ch

**Keywords:** burns, physiological basics, pathophysiological basics, wound healing, scar therapy, hypertrophic scar, dosage, connective tissue resistance

## Abstract

Deep dermal defects can result from burns, necrotizing fasciitis and severe soft tissue trauma. Physiological scar restriction during wound healing becomes increasingly relevant in proportion to the affected area. This massively restricts the general mobility of patients. External mechanical influences (activity or immobilization in everyday life) can lead to the formation of marked scar strands and adhesions. Overloading results in a renewed inflammatory reaction and thus in further restriction. Appropriate mechanical stimuli can have a positive influence on the scar tissue. “Use determines function,” and even minimal external forces are sufficient to cause functional alignment (mechanotransduction). The first and second remarkable increases in connective tissue resistance (R1 and R2) seem to be relevant clinical indications of adequate dosage in the proliferation and remodulation phase, making it possible to counteract potential overdosage in deep dermal defects. The current state of research does not allow a direct transfer to the clinical treatment of large scars. However, the continuous clinical implementation of study results with regard to the mechanosensitivity of isolated fibroblasts, and the constant adaptation of manual techniques, has nevertheless created an evidence-base for manual scar therapy. The manual dosages are adapted to tissue physiology and to respective wound healing phases. Clinical observations show improved mobility of the affected regions and fewer relapses into the inflammatory phase due to mechanical overload.

## 1. Introduction

Depending on the rehabilitation phase, therapies for burn injuries include pneumonia and contracture prophylaxis, promotion of mobility and independence, and strength endurance training. However, scar treatment is a particular focus for burns patients. Techniques and dosages are based on tissue physiology, wound healing phases and empirical clinical knowledge.

The fibres and cells which are responsible for high elasticity and mobility are mostly located in the dermis. For this reason, a loss of this layer leads to massive limitations (scar contractures). In addition, there are processes of wound healing which lead to a physiological restriction of the scar [1].

“Use determines function”—this is well known. However, the question of which dosage should be applied in which wound healing phase at which intervention time is not easy to answer. The following sections highlight possible answers based on clinical considerations and empirical experience.

## 2. Wound Healing Phases

The process of wound healing serves to restore the integrity of the skin as quickly as possible. This is essential for survival as the skin protects against infection and prevents fluid loss [2].

To make manual scar therapy effective, the physiological and pathophysiological processes of wound healing must be known. The aim is to apply physiological stimuli as adequately as possible in each phase of wound healing (functional stimulus). Within the wound healing phases, the tissue is mechanically more or less stable. Therefore, different dosages are required.

Wound healing can be divided into an inflammatory phase, a proliferation phase, and a remodulation phase. These phases are tissue-specific and differ accordingly. The literature mentions the following wound healing times:
-Inflammatory phase: until the 6th day-Proliferation phase: until day 21-Remodulation phase: from day 21. 

In patients with deep-dermal and large-area defects (burns or necrotizing fasciitis), the individual wound healing phases last significantly longer and are never homogeneous in themselves. Depending on the depth and size of the defect, scar areas are never in the same wound healing phase.

### 2.1. Inflammatory Phase

Vasodilation occurs in the surrounding uninjured tissue. As a result, the supply of the cells with nutrients and oxygen is optimized. A blood coagulum develops to ensure hemostasis. Macrophages begin to break down the destroyed tissue. Fibroblasts “scan” the surrounding tissue at the site of injury and begin to produce specific building materials. Among other things, macrophages and erythrocytes produce growth hormones, which stimulate the formation of granulation tissue and attract fibroblasts to the site of the injury (chemotaxis) [1]. The fibroblasts located on the surface differentiate in their contractile phenotypes, the myofibroblasts. Myofibroblasts try to close the wound. If they succeed, they die by apoptosis [3].

### 2.2. Proliferation Phase

This phase is characterized by the productive activity of fibroblasts and the ingrowth of vessels in the area of injury [1]. Fibroblasts play a central role in wound healing. They analyse the defective area, break down waste products and produce specific building materials for reconstruction. They also react very specifically to mechanical stimuli from outside. In this phase, fibroblasts begin to fill the wound with thin, unspecific and temporary type III collagen fibres. The turnover time is 30 days. During this period, an unformed connective tissue ball is formed [1].

In uninjured, physiologically stimulated tissue, the collagen fibres align themselves according to certain main pulling directions. Fibroblasts react very sensitively to mechanical stimuli from outside and align collagen fibres (or all types of fibres) according to the mechanical force in the Extra Cellular Matrix (ECM). If this functional stimulus does not occur in the proliferation phase, for example as a result of immobilization, fibroblasts cannot recognize any functional alignment and form a mechanically less stable collagen ball of unspecific collagen type III. If, however, a mechanically adequate stimulus is already provided in the proliferation phase, collagen type III will align itself functionally and according to physiological use (Figure 1). This is of utmost importance for manual scar therapy, since in the remodulation phase the definitive collagen type I always forms in the same orientation as the non-specific collagen type III [4,5]. Temporary collagen type III does not tolerate acceleration or shear forces. It is mechanically very unstable. Excessive mechanical stimuli tear the temporary structure and inevitably initiate a new inflammatory phase [4].

In the proliferation phase, the production of basic substance is very low in relation to collagen production. The necessary stability of the scar tissue is ensured in this phase by myofibroblasts. However, it is by no means comparable with the final stability of the scar tissue. It is therefore necessary to adjust the dosage of manual stimuli [3]. In healthy tissue, the production of basic substance generally predominates over the production of collagen fibres.

### 2.3. Remodulation Phase

In this phase, the production of the basic substance increases. The unspecific type III collagen fibres created in the proliferation phase are largely replaced by the definitive and much more stable type I collagen. As is well known, the temporary collagen type III is oriented towards mechanical stimuli from outside. Thus, functional stimulation in the proliferation phase is an important prerequisite for final stability and fibre orientation (functionality) in the remodulation phase. By converting collagen type III into type I, stability is increased and wound contracture is no longer necessary. As a result, myofibroblasts increasingly perish by apoptosis [3]. 

Collagen type I is mechanically very robust. Much greater (and longer-lasting) functional stimulation is required to effect adaptation. The turn-over time is 150 to 500 days, depending on the type of tissue. If no functional stimulation is provided during the proliferation phase, the definitive collagen type I—as described—will also form into a collagen ball. It will now require much longer-lasting and mechanically stronger stimulation before it is functionally aligned [1,4,5]. However, such remodelling is always accompanied by temporary tissue weakening. This must always be taken into account—especially in patients who are undergoing stress. Overdosage inevitably results in a new inflammatory phase, which must be avoided at all costs.

### 2.4. The role of Myofibroblasts

Fibroblasts located at the wound edges differentiate in a physiological way during wound healing in their contractile phenotypes, the myofibroblasts. Their task is to close the wound. Morphologically, myofibroblasts are characterised by a contractile “apparatus” of actin bundles which end in adhesion complexes on the surface of the myofibroblast. This system can transmit forces to the surrounding ECM and leads to wound closure. Myofibroblasts exhibit four times the contractility of normal fibroblasts [1].

After wound closure, myofibroblasts die as a result of apoptosis. This phenomenon cannot yet be fully explained. What is certain is that myofibroblasts do not undergo apoptosis in pathological scarring, thus creating a vicious circle. This was observed by Tomasek et al. in cases of delayed wound closure due to external factors (e.g., infection and mechanical overload) [3].

## 3. The Hypertrophic Scar

The hypertrophic scar is the most common pathological scar after deep dermal defects. In the following, the characteristics of a hypertrophic scar are in focus. The role of TGF-β 1 and the resulting therapeutic options will also receive attention.

A “hypertrophy” is always accompanied by enlargement of the tissue. Hypertrophic scars are very raised, but remain restricted to the actual wound area. They are recognisable by their redness, their elevation and their hypo- or hypersensitivity. Sometimes they can also react painfully to pressure and are associated with severe itching. It is possible for them to recede and flatten out over time.

Hypertrophic scars associated with burn wounds are associated with contractures that not only disfigure but also restrict function [6]. They occur more frequently after burns and delayed wound closure. Areas of the body with increased tension (e.g., in the area of the deltoid muscle or sternal area) and which allow a great deal of freedom of movement also tend to produce hypertrophic scarring (Figure 2) [6,7].

Although there are many studies on hypertrophic scars, understanding of their pathophysiology, and especially of therapeutic options, remains limited [6]. The reasons for this are manifold.

Research is often conducted with cell cultures (in vitro). Conclusions regarding humans are only possible to a limited extent. It is not clear to what extent the conditions are comparable.

Research on animal models (in vivo) comes much closer to the real conditions. It is questionable, however, whether the activity investigated is identical in humans and animals and whether research findings on wound healing in animals can be transferred to humans.

### The Role of TGF-β 1 in Hypertrophic Scars

TGF-β (Transforming Growth Factor Beta) is the classic example of a growth factor that is involved in many essential cell functions and processes via signal cascades inside and outside the cell. For example, growth factors are released after cell damage and trigger the wound healing cascade. They also play a key role in the differentiation of fibroblasts into myofibroblasts [3]. After injury, they can maintain pathologically occurring fibrosis [8]. The signalling process usually begins outside the cell (e.g., through a mechanical stimulus in the tissue). It is then carried on inside the cell via further signal proteins to the cell nucleus. There the transcription of certain genes increases or inhibition takes place (Figure 3). This leads to the expression of certain proteins. In the case of TGF- β 1 they influence various processes associated with wound healing [6,9,10]. Depending on the expression of the proteins, this can lead to an impact on the collagen network in the extracellular matrix which affects the mobility of the tissue.

The way TGF-β 1 works in cell cultures (in vitro) and animal models (in vivo) has already been extensively investigated. For example, it is known that there is a link between excessive wound healing and the formation of hypertrophic scars with an increased concentration of TGF-β 1.

In chronic and poorly healing wounds, there is a decrease in TGF-β 1, which is a further indication that growth factors play a significant role in wound healing [11]. However, neither direct clinical relevance nor a resulting therapeutic approach has been demonstrated.

## 4. Cell Biological Aspects

After a deep dermal defect, the goal of therapeutic treatment is always to restore function at the structural and activity level (ICF). This is the prerequisite for the patient to be able to participate in life again.

In a first step, this usually requires surgery (e.g., split skin covering). Immobilization of the affected structures is necessary until the tissue has healed. As soon as the surgical procedure allows this, the therapist can apply functional stimuli (e.g., manual scar therapy) to the tissue. With this treatment, the affected structure (i.e., the scar) can be positively influenced at the cellular level, including the extracellular matrix (mechanotransduction). These functional mechanical stimuli are eliminated by postoperative immobilization or as a result of limitations caused by very fragile skin conditions. It is known from the literature that this absence is associated with a loss of functional alignment of the fibres. It has also been proven that too much mechanical stress quickly leads to an overdose (causing cell damage and a renewed inflammatory reaction) [1,4,5,8,11,12,13].

The transmission of mechanical stimuli in the body by, e.g., manual techniques, appears to take place in two signalling pathways. Empirically, a “release” in the tissue can often be perceived after only a few minutes. The mechanical stimulus must therefore have an effect directly in the tissue (for example, the release of possible pathological crosslinks or reactive tone changes in the connective tissue).

The second signalling pathway has been scientifically researched under the term “mechanotransduction”. This is the reaction inside the cell to a mechanically applied stimulus from outside. If the stimulus is adequate, the cell reacts with gene transcription and thus influences the cytoskeleton and the ECM and thus the quality of the affected tissue [8,12,14,15,16,17,18].

The aim of the therapeutically applied manual techniques is therefore not (as previously assumed) to lengthen collagen fibres by stretching, but to directly influence the cell biological processes by means of adequate stimulation.

### 4.1. How Do Fibroblasts Perceive a Mechanical Stimulus from Outside?

Recent investigations have shown that tiny cilia have an important function in the registration of shear forces [19]. These hairs are able to register the “fluid shear” of the surrounding basic substance in order to transmit corresponding information to the cell interior. They are very sensitive and require only gentle mechanical impulse forces to become active.

### 4.2. Which Mechanical Stimuli Cause Which Fibroblast Activity?

If the structure is not injured, a constantly recurring 4% elongation (e.g., of a ligament) is required in order for the fibroblast to adapt the collagen structure and the matrix. Highly dosed and jerky tensile loading cause the fibroblast to release proinflammatory messenger substances. This may initially promote wound healing, but with repetitive application it may result in stagnating wound healing dynamics [20].

Immobilisation lasting several weeks or chronic lack of exercise, on the other hand, leads to the formation of additional crosslinks. The collagen fibres lose their natural wave structure (a process known as “crimp”) [20].

Carano et al. (1996) conducted research on cell cultures of artificial ligaments. In doing so, they focused the stretching stimulus on the fibroblast and measured the effect on wound healing. A comparison of stretching stimuli (3%–12%) and application times (1–5 min) showed that a 3% stretching load and a five-minute application time were most effective in accelerating the healing of a previously placed lesion [21].

Zein-Hammoud et al. (2015) exposed fibroblasts in a cell culture to a constant basic voltage. A one-time manual voltage reduction lasting sixty seconds reduced the proinflammatory consequences of a previous repetitive motion-injury (RSI) simulation. By stimulating a myofascial release technique (low-grade tissue stretching, 60 s), there was also a muted inflammatory response following RSI stimulation - in addition to the reduced apoptotic effect of this predamage [22].

Furthermore, the authors were able to demonstrate that fibroblasts exposed to a low-grade fluid shear in cell culture reacted with increased expression of the collagen-degrading enzyme MMP-1 in a time window of four to eight hours [23].

### 4.3. Mechanotransduction

Figure 4 shows the mechanotransduction process schematically. The mechano-sensors are stimulated by a mechanical stimulus from outside (here using the example of a fibroblast). These in turn activate “adapter proteins”, which are located on the cell nucleus membrane (nuclear membrane). These “adapter proteins” transmit the mechanical stimulus to the cytoskeleton. In this way they stimulate the skeletal structure. The cell reacts with gene transcription and passes on correspondingly altered (adapted) processes and products to the extracellular matrix. Thus, a cell response to a mechanical stimulus from outside takes place [8,24].

The cytoskeleton of every cell is normally under physiological tension. This tension is arranged in balance with the consistency of the ECM and the presence of functional mechanical stimuli in everyday life.

In the case of pathological scarring, however, physiological tension in the cytoskeleton is already increased. This leads to increased sensitivity to mechanical stimuli. Due to the increased storage of collagen and the activity of myofibroblasts, the ECMT has a higher tension and stiffness than physiological skin (tissue). This imbalance is responsible for the fact that mechanical stimuli related to everyday life can already be perceived as overload at the cellular level and lead to an overreaction in the cell. This can end in a pathological fibrosis process.

### 4.4. Clinical Conclusion

In order to be able to initiate the correct processes at the cellular level, adequate manual dosing is a basic requirement. Overloading the tissue inevitably ends in cellular damage and triggers a new inflammatory reaction with all the cardinal symptoms. Underloading, on the other hand, is manifested in the form of crosslinks and reduced elasticity and resilience. In the follow-up treatment of deep dermal defects, additional parameters must be taken into account.

## 5. Increase in First and Second Connective Tissue Resistance (R1 and R2)

Basically, it is important to distinguish between the quantity and quality of movement. The quantity of movement is the physically objectively measurable extent of the movement. In contrast, the quality of movement includes the flow of movement, dynamics, rhythm and harmony of the movement. These are qualities that can be subjectively felt during movement. In order to be able to assess the quantity and quality of a movement properly, the therapist first needs theoretical knowledge: what quantity and quality can be expected of a joint or tissue in an uninjured state? In addition, however, rich practical experience and sensitivity are also required.

As a rule, a healthy joint or tissue always behaves in the same way: it has a smaller or larger “neutral zone” within its range of motion. At the end of the range of motion there is a “physiological space”, followed by a “paraphysiological space” (Figure 5).

The “neutral zone” is usually located in the middle of the range of motion and is characterised by a very small increase in resistance. By achieving the second remarkable increase in the resistance of the connective tissue the “physiological space” begins. The therapist passively mobilises the tissue more or less far into this area (depending on the intensity of the manual grading (dosage) and the currently prevailing wound healing phase). The “paraphysiological space” can only be reached by impulse mobilisation (manipulation). The anatomical barrier is moved forward post-traumatically or post-operatively. The reason for this is collagen type III, which predominates in the proliferation phase. This is responsible for the integrity of the injured structure and causes the rapid formation of water-soluble crosslinks. Temporary collagen type III does not withstand mechanical shear and acceleration forces very well. Overdosage is therefore quickly possible, particularly in the proliferation phase. Microtraumatic injuries trigger new phases of inflammation, which can lead to increased tissue restriction, thus restricting movement.

## 6. Dosage in Manual Scar Therapy

Unfortunately, the facts explained so far do not allow for a clear conclusion regarding the correct dosage. However, by putting the individual pieces of the puzzle together, it is possible to approach the issue of adequate and functional dosage.

It can be assumed that there is a big difference between the application of a high and a low dosage. This consideration applies to the active phase of a scar and its not-yet-mature state. Therefore, amplitude, duration and frequency are important parameters that should be considered in connection with wound healing phase [8].

### 6.1. Amplitude

Amplitude is defined as the largest excursion of an oscillation or pendulum from its centre position or as the amplitude of the oscillation. If the mechanical stimulus (the amplitude) is too strong, the reaction of the cell nucleus becomes much less controllable. An overreaction of the cell can quickly occur. Due to the persistent inflammatory state, the cell is more sensitive. This leads to the hypothesis that stimuli which are harmless for the physiological skin may already represent an overdose in scars [26].

Since mechanotransductive cell responses are already initiated by minimal forces (up to 0.00058 N), functional alignment is already achieved with very gentle manual therapeutic interventions [14]. The empirically known remarkable increases in connective tissue resistance (R1 and R2) in the back area are 1–2 N for R1 and 2–4 N for R2 [27,28]. It can therefore be assumed that manual interventions in the area of R1 probably already provide an adequate stimulus for the scar tissue to produce a mechanotransductive response of the cell and thus a functional alignment of the extracellular matrix.

In the proliferation phase, R1 appears to be an important indication of a wound-healing-adapted dosage. In the remodulation phase, a dosage up to R2 is possible.

Several years of clinical experience with severely burn-injured patients show that manual scarring techniques in the proliferation phase should only be used until the first increase in connective tissue resistance. This is because the tissue is significantly less resilient than comparable connective tissue in the same wound healing phase [29]. Because of the delayed wound healing, burn scars often have a strongly prolonged proliferation phase. Thus, the transition from R1 to R2 is empirically experienced as smooth.

Koller conducted a pilot study on inter-rater reliability to determine R1 and R2 [27]. The results were promising. Therapists seem to be able to detect the first significant increase in connective tissue resistance with moderate inter-rater reliability (ICC2=0.67) and the second significant increase in connective tissue resistance with good inter-rater reliability (ICC2=0.80). With regard to direct and indirect pain conduction, it can be assumed that C-fibre activity starts increasing with the second marked increase in connective tissue resistance (R2), and that up to R2 mainly the Aδ fibre is active as a “warning signal” of cell damage [27,28].

Figure 6 shows the specific dosage for deep dermal defects. The dosage is significantly lower due to the prolonged wound healing phase associated with this type of injury.

Figure 7 shows the dosage adapted to the wound healing phase in manual therapy for general injuries and operations with normal tissue-specific wound healing.

### 6.2. Frequency

The frequency describes how quickly repetitions within a periodic process follow each other for one second. The unit of frequency is Hertz (Hz).

Balestrini and Biliar found that a cyclical strain stimulus with 0.2 Hz and moderate amplitude stimulates the fibroblasts to produce a more resistant matrix. The tissue density increased, and a reorganisation of the fibres takes place. Histologically, a great decrease in thickness was observed. After this stimulus, the tissue was thinner, denser and better organized [13].

Carano et al. demonstrated a 200% increase in collagenase production with cyclic stretching on fibroblasts compared to static stretching [21].

Clinically recommended would therefore be a slight stretching, adapted to the respective wound healing phase and with intermittent oscillation (0.2Hz) at the end of the respective amplitude. These assumptions correlate with empirical and clinical experience, but have not yet been proven in human research.

### 6.3. Duration

How long should a stimulus be applied? This is another fundamental question regarding the dosage. Although splints and other stretching techniques are widely used, there are no human test results available in the literature. Only effects from animal studies can serve as a basis for clinical considerations and empirical experience.

Bouffard et al. have already demonstrated good results with a daily ten-minute stretch and moderate amplitude (20% stretch) (reduced TFG β 1) [8]. However, practical experience shows that splint applications over several hours also produce positive effects.

Carano et al. (1996) conducted research on cell cultures of artificial ligaments. They found that a five-minute application time was most effective in terms of wound healing of a previously placed lesion [21].

Empirically, good results can be achieved with application times of one minute per localisation, three to five times per therapy unit. As a rule, one to two therapy units per day are realistic in the clinical setting.

## 7. Dosage Recommendations

⇨In the proliferation phase: in the area of the first remarkable increase in connective tissue resistance (amplitude).⇨In the remodulation phase: increasing until the second remarkable increase in connective tissue resistance (amplitude).⇨Application duration in both phases: usually one minute per localization, three to five times per therapy unit, supplemented by an oscillating frequency of 0.2 Hz at the end of the respective amplitude.

## Figures and Tables

**Figure 1 ijms-21-02055-f001:**
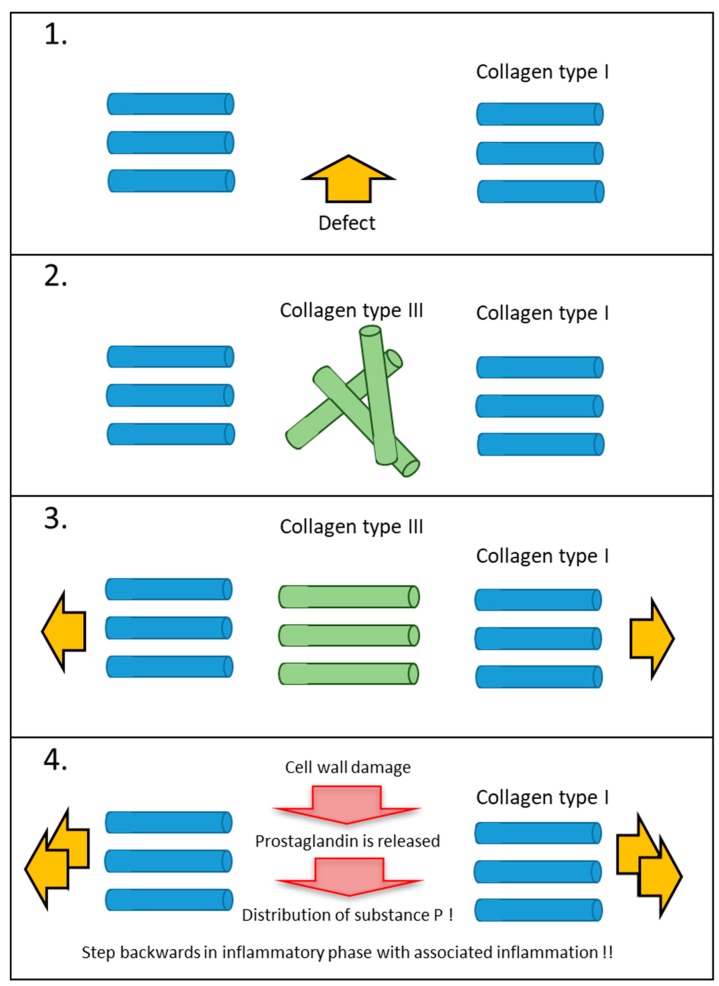
Cellular reactions to differently dosed mechanical stimuli (tissue or scar tissue). (Source: own illustration). (1) A deep dermal injury causes a defective site in the collagen network. The matrix is torn at this point and physiological repair processes in the form of inflammation are set in motion. (2) In the proliferation phase, the fibroblast synthesizes collagen type III. Without a mechanical stimulus from outside, this reticular collagen is not functionally aligned in the extracellular matrix. (3) With adequate and functional external mechanical stimulation, the reticular collagen is aligned according to function. This has a decisive advantage, because in the remodulation phase the definitive collagen type I is aligned in exactly the same way as the existing reticular collagen. (4) If too much mechanical stimulation is applied during the proliferation or remodulation phase, renewed damage occurs at the cell level. A new, unwanted inflammatory reaction with increased tissue restriction occurs.

**Figure 2 ijms-21-02055-f002:**
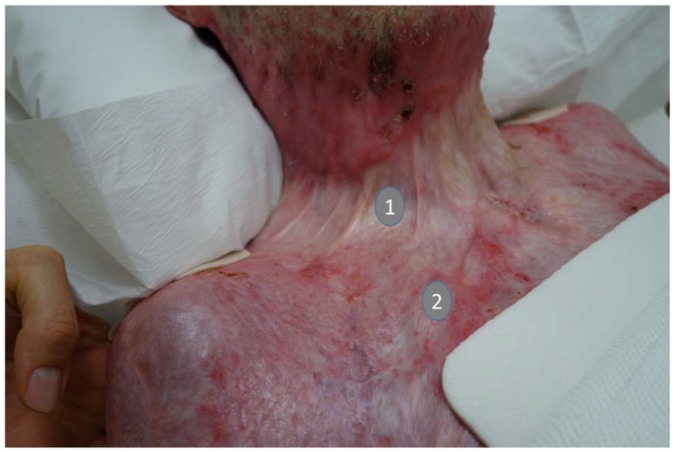
Typical course of scar contractures in the neck area. Legend: (1). Concavely scarred strand in the area of the neck. The scarred strands are mechanically stable and transmit mechanical forces during movements without damage. (2). Scar areas that are still in the inflammatory or proliferation phase (red scar areas) react to mechanical forces that are too strong with a tension equalization and aninflammatory reaction. Therefore, in manual scar therapy, attention must always be paid to how the mechanical stimulus is applied within the scar area. If necessary, fragile scar areas must be mechanically protected by hand. (Photo: Property of the Bellikon Rehabilitation Clinic).

**Figure 3 ijms-21-02055-f003:**
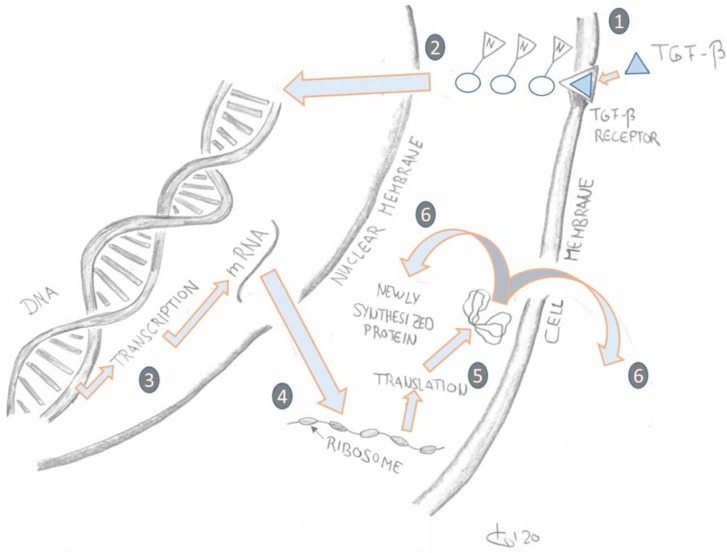
Schematic simplified representation of Transforming Growth Factor Beta (TGF-β) signal transduction outside and inside the cell [9,10]. (Graphic: T. Koller.) Legend: (1). TGF-β attaches to the TGF-β receptor on the cell membrane. (2&3). Signalling transcription factors transduce the extracellular TGF-β signalling from the cell-bound TGF-β receptors into the nucleus where they activate or inhibit transcription of TGF-β target genes into the respective mRNA sequences. (4&5). In the cytoplasm the mRNA sequences are translated by ribosomes into proteins. (6). The newly synthesized proteins are either exported out of the cell to form collagen or they are used for intracellular processes.

**Figure 4 ijms-21-02055-f004:**
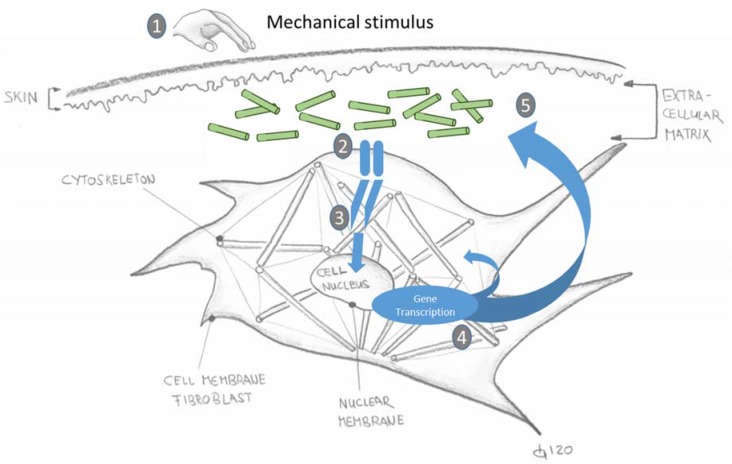
Schematically depicted mechanotransduction process of the reaction of a fibroblast cell to an external mechanical stimulus [8,12,14,15,16,17,18] (simplified graphic: T. Koller). Legend: 1. A mechanical stimulus is applied by a physiological movement or manual input by hand. (2). Mechanosensors on the cell membrane register the fluid displacement and transmit this signal into the cell interior. (3). The mechanosensors activate “adapter proteins” which transmit the mechanical stimulus to the cytoskeleton. (4). The cell reacts with gene transcription and passes on correspondingly altered (adapted) products to the extracellular matrix. (5). Newly functionally aligned collagen networks after an injury or through “new use” due to a new activity.

**Figure 5 ijms-21-02055-f005:**
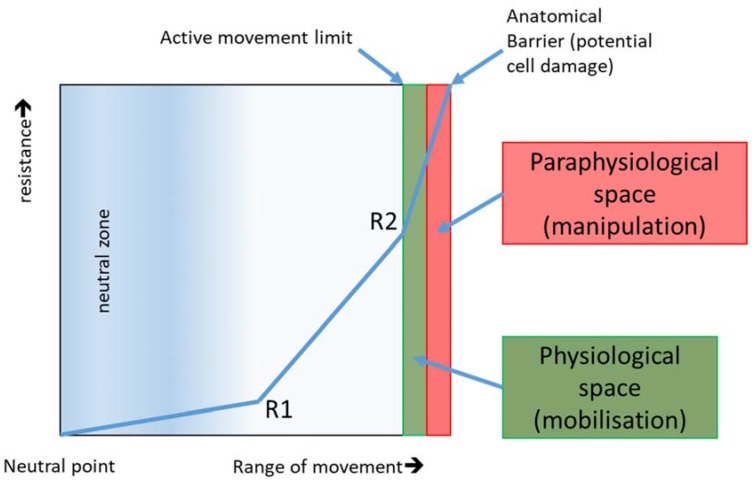
Behaviour of resistance depending on the extent of movement in joints and connective tissue. R1 represents the first and R2 represents the second remarkable increase in connective tissue resistance. Depending on the intensity of the manual degree classification (dosage), therapists treat passively more or less into physiological space. The paraphysiological space is only achieved by impulse mobilisation. If the anatomical barrier is crossed, cell damage occurs (lesion) and reacts with an inflammatory reaction. This anatomical barrier is important in post-traumatic and postoperative conditions due to adhesions and water soluble crosslinks to the front postponed. Especially in the proliferation phase, it can quickly lead to an overdose. The microtraumatic injuries trigger new inflammatory phases, which result in increased restriction of movement and restriction [25] (source: own illustration).

**Figure 6 ijms-21-02055-f006:**
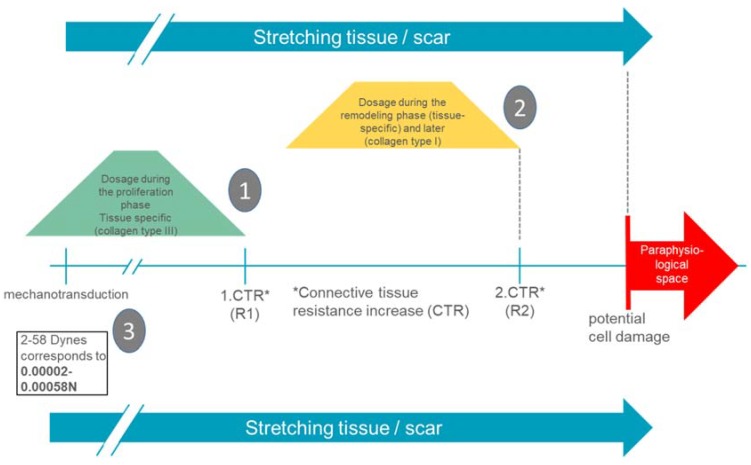
Summarizing presentation of wound-healing-phase-adapted and tissue-specific dosages in the manual therapy of burn scars (source: own presentation). Legend: (1). In the proliferation phase, in the area of the first remarkable increase in connective tissue resistance (amplitude). (2). In the remodulation phase, increasing until the second remarkable increase in connective tissue resistance (amplitude). (3). R1 and R2 are exemplary. Connective tissue displaceability in the back is located in a range of 1–4 N [28] (much higher than the fibroblast needs for mechanotransductive cell response).

**Figure 7 ijms-21-02055-f007:**
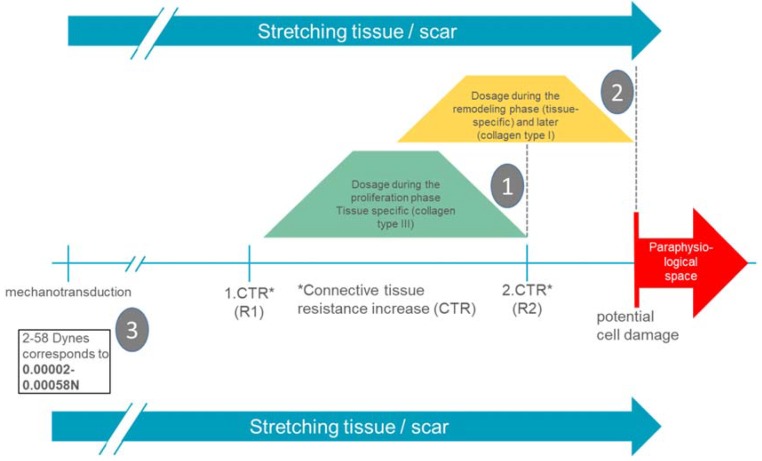
Summarizing presentation of wound-healing phase adapted and tissue-specific dosages in manual therapy of injured tissue (post-traumatic/postoperative) (source: own presentation). Legend: (1). In the proliferation phase, increasing until the second remarkable increase in connective tissue resistance (amplitude). (2). In the remodulation phase, in the second remarkable increase in connective tissue resistance (amplitude). (3).R1 and R2 are exemplary. Connective tissue displaceability on the back is located in a range of 1–4 N [28] (much higher than the fibroblast needs for a mechanotransductive cell response).

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
