# Peer review of "Mechanosensitive Aspects of Cell Biology in Manual Scar Therapy for Deep Dermal Defects"

_ijms, 2020, doi:10.3390/ijms21062055_

Round 1

Reviewer 1 Report

The article is easy to follow for a reader and gives you a feel of a book chapter.

However, I would recommend improving figure presentation specially figure 1. Hand drawn figures are good. If possible try redrawing them in a software with color scheme. Can be done easily. 

The abstract could be expanded further, giving a line or two about what does the article talks about and not just some background information left as is. Right now the abstract is left hanging for the reader to go through the complete article to find out what useful information can be found.

Additionally, review for editorial and spell check in the article.

Author Response

Dear Reviewer

attached the replies of the first revision.

all text changes are written in green.

Kind regards

- Are the methods sufficiently described? Not available:

         As I have collected the literature over years, I cannot write anything about the method. I have addressed this with the publisher mdpi Verlag. The publisher said that I have to describe the method part of an article or a systematic review. If there is no methods section available, it will be a normal review.

         Correction: page 1, line 1: article replaced by review.

         Article Review

- Colour illustrations:

         Correction: Figures 1, 3 and 4 have been revised in colour. The PowerPoint files are available to the publisher for possible colour adjustments.

- Summary too short:

         I have extended the summary and better illustrated the topic of the review.

         Correction: Addition page 1, lines 19-25.

                   The current state of research does not allow a direct transfer to the clinical treatment of large scars. However, the continuous clinical implementation of study results with regard to the mechanosensitivity of isolated fibroblasts and the constant adaptation of manual techniques has nevertheless created an evidence-based basis for manual scar therapy. The manual dosages are adapted to the tissue physiology and the respective wound healing phase. Clinical observations show improved mobility of the affected regions and fewer relapses into the inflammatory phase due to mechanical overload.

All new PowerPoint files are available to the publisher.

Reviewer 2 Report

This study is interesting, but several points should be clarified are needed prior to publication.

Major concerns:

Need more description / explanation for All Fig

Author Response

Dear Reviewer

attached the replies of the first revision.

all text changes are written in green

Kind regards

- Requires more description / explanation for all illustrations:

         Correction:

                  Figure 1: Deleted page 4, lines 4-11.

  1. defective site in the collagen network.
  2. synthesis of collagen type III in the proliferation phase

        without external mechanical stimulus.

  1. synthesis of collagen type III in the proliferation phase with                      adequate functional, mechanical stimulus from outside
  2. reaction to an excessive mechanical stimulus in the                                  proliferation phase (also possible in the remodulation phase;                      thus a new inflammatory phase through cell wall damage).

                  Addition page 3, line 4 to page 4, line 3.

  1. A deep dermal injury causes a defective site in the collagen network. The matrix is torn at this point and the physiological repair processes in the form of inflammation are set in motion.
  2. In the proliferation phase, the fibroblast synthesizes collagen type III.

          Without mechanical stimulus from outside, this reticular collagen is not              functionally aligned in the extracellular matrix.

  1. with adequate and functional external mechanical stimulus, the reticular collagen is aligned according to function. This has a decisive advantage, because in the remodulation phase the definitive collagen type I is aligned exactly the same way as the existing reticular collagen.
  2. If too much mechanical stimulation is applied during the proliferation or remodulation phase, renewed damage occurs at the cell level. A new, unwanted inflammatory reaction with increased tissue restriction occurs.

         Figure 2: Addition page 5, lines 18-24.

        Legend: 1. Concavely scarred strand in the area of the neck. The scarred                        strands are mechanically stable and transmit the mechanical                               forces during movements without damage.

                  2. Scar areas, which are still in the inflammatory or proliferation                        phase (red scar areas), react to mechanical forces that are too                          strong with a tension equalization and an inflammatory reaction.                        Therefore, in manual scar therapy, attention must always be                              paid to how the mechanical stimulus is applied within the scar                            area.  If necessary, fragile scar areas must be mechanically                              protected by hand.

         Figure 3: Addition page 6, lines 15-22

                      Legend:
                                      1. TGF-β attaches to the TGF-β receptor on the cell                                          membrane.
                                     2&3. Signaling transcription factors transduce the                                             extracellular TGF-β signaling from the cell membrane                                       bound TGF-β receptors into the nucleus where they                                           activate or inhibit transcription of TGF-β target genes                                       into the respective mRNA sequences.
                                     4&5. In the cytoplasm the mRNA sequences are                                               translated by ribosomes into proteins.
                                     6. The newly synthesized proteins are either exported                                      out of the cell to form collagen or they are used for                                          intracellular processes.

         Figure 4: Addition page 9, lines 1-9.

                      Legend: 1. A mechanical stimulus is applied by a physiological                          movement or manual input by hand.

  1. Mechanosensors on the cell membrane register the fluid displacement and transmit this signal into the cell interior.                    
  2. The mechanosensors activate "adapter proteins" which transmit the mechanical stimulus to the cytoskeleton.
  3. The cell reacts with gene transcription and passes on correspondingly altered (adapted) processes and products to the extracellular matrix.
  4. Newly functionally aligned collagen network after an injury or through "new use" due to a new activity.

         Figure 5: Deleted page 10, lines 2-4.

                   Resistance as a function of the extent of movement in joints and connective tissue (physiological). R1 represents the first significant increase in    connective tissue resistance, R2 the second significant increase in connective       tissue resistance

         Addition page 10, lines 5-13

        Behaviour of resistance depending on the extent of movement in                        joints and connective tissue. R1 represents the first and R2 represents                the second remarkable increase in connective tissue resistance.                        Depending on the intensity of the manual Degree classification                          (dosage) therapists treat passively more or less into physiological                      space. The paraphysiological Space is only achieved by impulse                          mobilisation. If the anatomical barrier is crossed, cell damage occurs                   (lesion) and reacts with an inflammatory reaction. This anatomical                     barrier is important in post-traumatic and postoperative conditions                     due to adhesions and water soluble crosslinks to the front postponed.                 Especially in the proliferation phase, it can quickly lead to an                             overdose. The microtraumatic injuries trigger new inflammatory                         phases, which result in increased restriction of movement and                           restriction

         Figure 6: Addition page 11, line 30 to page 12 ,line 5

                      Legend: 1. In the proliferation phase, in the area of the first                 remarkable increase in connective tissue resistance (amplitude).

  1. In the remodulation phase, increasingly until the second remarkable increase in connective tissue resistance (amplitude).
  2. R1 and R2 are exemplary connective tissue displaceability on the back is located in a range of 1-4 N [28]. So much higher than the fibroblast needs for a mechanotransductive cell response.

         Figure 7: Addition page 12, lines 9-15.

                      Legend: 1. In the proliferation phase, increasingly until the                  second remarkable increase in connective tissue resistance (amplitude).

  1. In the remodulation phase, in the second remarkable increase in connective tissue resistance (amplitude).
  2. R1 and R2 are exemplary connective tissue displaceability on the back is located in a range of 1-4 N [28]. So much higher than the fibroblast needs for a mechanotransductive cell response.

     Text Addition page 6, lines 9-11

     Depending on the expression of the proteins, this can lead to an impact on         the collagen network in the extracellular matrix which affects the mobility of       the tissue.

         All new PowerPoint files are available to the publisher.